# OpenReview forum: "New Wide-Net-Casting Jailbreak Attacks Risk Large Models"
_ICML.cc/2026/Conference — ICML 2026 regular_

### Official Review · Reviewer_yXcy · 2026-03-10

**Soundness:** 3
**Presentation:** 2
**Significance:** 3
**Originality:** 2
**Overall Recommendation:** 3
**Confidence:** 4

**Summary:**

This work studies a previously underexplored wide-net-casting jailbreak setting, where an attacker queries a group of large models and succeeds if any one model returns harmful content. It first shows that directly adapting existing single-model jailbreak methods into the wide-net-casting setting substantially increases ASR and toxicity across LLMs and MLLMs, including same-family targets and targets with safeguards. It then proposes a tailored model-based attack that jointly trains specialized generators for different targets and achieves stronger results.

**Compliance With Llm Reviewing Policy:**

Affirmed.

**Final Justification:**

After taking all factors into consideration, I've decided to maintain my rating. While interesting, the work's design and novelty are not at an acceptable level.

**Key Questions For Authors:**

Please see the weaknesses. I appreciate the practical evaluation scenario in this work, and I tend to raise my score if these questions can be addressed well.

**Limitations:**

Yes.

**Strengths And Weaknesses:**

- Pros:
  - This work identifies a realistic and important threat model that is genuinely different from standard single-model jailbreak evaluation, in which attackers can query multiple models and only need one failure. The WASR framing captures this risk.
  - The tailored attack is more than a simple heuristic ensemble.
- Cons:
  - The core observation that attacking multiple independent models with OR-aggregation increases success probability is statistically trivial. This work would benefit from a more explicit discussion of what non-obvious insights emerge beyond this baseline probabilistic amplification, and whether the observed WASR gains exceed what independent-OR probability would predict. Theoretically, I only need to mix a weak model into some strong models to achieve a very high MASR. Besides, the evaluation relies on a binary ASR judge and a single toxicity score to assess jailbreak outputs. Neither metric captures the specificity, actionability, or informativeness of harmful responses, which are arguably more relevant to real-world threat severity than a coarse success/toxicity signal.
  - The methodological novelty is moderate. The scenario itself is novel and valuable, but the tailored method can also be viewed as an entropy-regularized specialization schedule over multiple generators, rather than a fundamentally new jailbreak paradigm.
  - The evaluation breadth is insufficient. Most experiments use a fixed group size. Closed-source evaluation is only for MLLMs.
  - It reports averages over five runs, but the main tables do not show variance. Since some gains over naive strategies are not huge, uncertainty estimates would strengthen the empirical case.
  - The specialization training and all evaluations use train–test splits drawn from the same dataset (AdvBench or MM-SafetyBench). No cross-dataset generalization experiment is conducted, leaving it unclear whether specialized generators can handle out-of-distribution harmful intents whose topics or phrasing diverge substantially from the training distribution.
  - The target LLMs are primarily 7–9B models, with some notably dated (e.g., Vicuna-7b-v1.5), and the MLLMs are similarly small. Evaluating on larger, more recent models with stronger safety alignment (e.g., Llama-3-70B, Qwen3, and GPT-5.X) would be necessary to substantiate the claimed real-world threat level.

---

> ### Author Rebuttal · Authors · 2026-03-31
>
> >*Q1: OR-aggregation [...] non-obvious insights [...] whether the observed WASR gains exceed independent-OR probability [...] mix a weak model into some strong models.*
>
> **A1: (1)** Jailbreak risk has direct societal impacts, as successful attacks may induce very harmful outputs like fraud, violence, self-harm. Our paper shows **OR-aggregation** can greatly increase risk, e.g., in 4th row of results in Tab 1, **mixing a weaker model into stronger ones leads to WASR exceeding ASR of any model (even the weak model itself)** by at least 23%. Given the tight link between jailbreak risk and real-world harm, revealing this previously undisclosed risk is itself important.
>
> **(2)** Our contribution goes beyond the OR effect. Once this setting is recognized, skilled adversary can design tailored attacks, achieving gains far beyond independent-OR baseline. We cast the problem as an exploration-exploitation problem and derive a closed-form solution via infinitesimal value shifting, Lagrange multipliers, and KKT conditions, which serves as a theoretically grounded generator update scheduler. With our proposed tailored scheduler, **WASR far exceeds independent-OR aggregation prediction, even reach 100% in some cases. This is thus a non-obvious conclusion**.
>
> **(3)** We also tested alternative OR-based variants, e.g., independent OR-aggregation of multiple attack queries on the same large model, and independent OR-aggregation of attacks from multiple generators against the same large model. We observe our WASR still significantly exceeds these OR-based probability baselines (variants) by more than 20%  (see A2 of Reviewer Be44 for more details).
> >*Q2: ASR and toxicity score.*
>
> **A2:** Prior jailbreak works (Wang et al., 2025; Miao et al., 2025; Zou et al., 2023) use ASR and toxicity score, which we follow. As suggested, we further evaluate response quality (**actionability** and **informativeness**) on AdvBench, via 5 LLM judges and a 5-human participant user study. Response quality is rated on a 1-5 scale (5 highest, 1 means jailbreak fails).
>
> ||Avg Score (LLMs)|Avg Score (Humans)|
> |-|-|-|
> |Baseline(Remiss)|2.5|2.3|
> |Naive Strategy 1|3.0|2.8|
> |Naive Strategy 2|3.1|3.0|
> |**Ours**|**4.1**|**4.0**|
>
> Our method consistently outperforms others.
> >*Q3: Methodological novelty.*
>
> **A3:** To perform effective attacks in our identified novel and valuable scenario, we propose a novel, tailored method to properly assign updates across adversarial sample generators.
>
> While our method involves an entropy-regularized formulation, this formulation is newly derived and specifically developed for this problem. Reaching this formulation and deriving how to assign updates from it is challenging. Using infinitesimal value shifting, Lagrange multipliers, KKT conditions, and complementary slackness, we first formalize this new problem and solve it, finally yield an elegant closed-form solution due to our careful and new theoretical derivations. This solution enables us to effectively perform tailored attacks in our novel scenario, and makes our method new. Via this method, we significantly outperform compared methods, achieving jailbreak success rates of even up to 100% in some settings (Tabs 5–6).
> >*Q4: Fixed group size [...] Closed-source evaluation is only for MLLMs.*
>
> **A4:** Besides fixed **group size** $M=4$ in main paper, we have tested other group sizes, e.g., $M=2,3$ in Tab 8 of Supp. We here further extend to $M=6$ (4 large models in Tab 2, LLaVA-1.6-vicuna-13b, and LLaVA-1.6-vicuna-7b) and $M=8$ (further add LLaVA-llama2-13b and LLaVA-llama2-7b), achieving WASR of 95.4 and 97.1. Results consistently show larger groups lead to higher risk.
>
> We also test on closed-source LLMs (GPT-3.5-Turbo-1106, GPT-4-0613, GLM-4-Plus, GLM-4-Long, GLM-4.7, Qwen-Plus, Qwen3-Max, and Qwen-Long). **Our method exposes substantial risks on closed-source LLMs**, getting WASR of 82.7, outperforming Baseline-ReMiss (68.3), Strategy 1 (70.9), Strategy 2 (71.5). We'll add detailed results to paper.
> >*Q5: Uncertainty estimates.*
>
> **A5:** We significantly outperform other methods, with statistically significant improvements (see A3 of Reviewer mgDS for more details). We'll include uncertainty estimates (STD) in all tables in paper.
> >*Q6: Cross-dataset experiment.*
>
> **A6:** Training on AdvBench and testing on MM-SafetyBench, we achieve 90.6 WASR, much higher than Baseline-MLAI+PixArt-α (79.1), Strategy 1 (80.8), Strategy 2 (81.5). Training on MM-SafetyBench and testing on AdvBench, we achieve 85.2, still outperforming Baseline (72.3), Strategy 1 (74.1), Strategy 2 (74.8).
> >*Q7: Evaluating on suggested larger models.*
>
> **A7:** We evaluate on suggested models: On LLMs (**Llama-3-70B**, **Qwen3**), we obtain 39.8 WASR, outperforming Baseline-ReMiss (26.1), Strategy 1 (28.4), Strategy 2 (29.1); On MLLMs (**GPT-5.1**, Qwen-VL-Max), we get 43.1, outperforming Baseline-MLAI+PixArt-α (31.9), Strategy 1 (33.7), Strategy 2 (34.5). We'll add detailed results to paper.

---

> > ### Author Rebuttal · Reviewer_yXcy · 2026-04-01
> >
> > Some concerns remain (and have even been amplified to some extent).
> > - A1: This only partially addresses my concern. It does not cleanly disentangle the paper's claimed insight from the basic OR-aggregation/weakest-link effect, and the additional comparisons are not the same as the independent-OR baseline I originally asked about.
> > - A3: This does not fully resolve my concern. The rebuttal reiterates that the method is formally derived and effective, but it does not change my view that the methodological novelty remains moderate relative to the novelty of the threat setting itself.
> > - A7: This directly addresses my previous concern, **but the new results also suggest that the strongest alarming claims in the paper may be driven largely by weaker targets.** On stronger contemporary models, the absolute WASR drops substantially, so the practical threat severity appears much lower than the manuscript suggests.
> > - Thank you very much for preparing the reply. Other concerns are mostly addressed.

---

> > > ### Author Response · Authors · 2026-04-03
> > >
> > > >*Further question about A1.*
> > >
> > > **Answer:** Thank you for mentioning the basic OR-aggregation/weakest-link effect. Our method can significantly improve attack success rate compared to these effects.
> > >
> > > For basic OR-aggregation effect, we clarify that we have compared extensively against it in both our paper and our previous responses. Specifically, **the basic OR-aggregation corresponds exactly to what we denoted as "baseline" in paper and in our previous responses**. As described in L372-374 & 140-151, this "baseline" independently attacks multiple target models and counts a case as successful if any model is breached. This is exactly what basic OR-aggregation is doing.
> > >
> > > As shown in paper (e.g., Tabs 5-7 & 14-17) and in our previous responses, **across extensive experiments, WASR gain of our method consistently and significantly exceeds that of basic OR-aggregation (which was called "baseline"). This shows that risk introduced by our method is substantially higher than risk introduced by basic OR-aggregation**. To avoid confusion, we'll replace term "baseline" to "baseline (basic OR-aggregation)" to make it clearer.
> > >
> > > We also apologize for the earlier misunderstanding. Since we had compared extensively with basic OR-aggregation (which was called "baseline"), we thought earlier you were asking about a different comparison.
> > >
> > > Meanwhile, thank you for mentioning weakest-link effect. As shown in paper, when one large model is obviously weaker than the others (see Tab 1), resulting in a clear weakest-link effect, our method still far outperforms both the weakest model and basic OR-aggregation (which was called "baseline") (see Tabs 1&5). Below, we conduct a further experiment over this effect, on 4 models. Among them, 3 models are strong ones equipped with the additional safeguard ASTRA, resulting in relatively lower individual ASRs, including Qwen2.5-VL-7B (9.4), MiniGPT-4-13B (10.9), & InstructBLIP-vicuna-13B (12.8), while LLaVA-1.6-Vicuna-7B uses only its original safety alignment and thus has a relatively higher ASR of 37.6, making it the weakest link among the 4 models. On these 4 models, even when a clear weakest-link effect exists, our method achieves 88.1 attack success rate, far higher than both weakest model result (37.6) and "baseline (basic OR-aggregation)" result (58.9). This further distinguishes our insight from basic OR-aggregation/weakest-link effect.
> > >
> > > >*Further question about A3.*
> > >
> > > **Answer:** Thank you for acknowledging that our threat setting (scenario) is novel and valuable. Our work identifies this important setting for the first time and proposes a tailored attack method to expose its safety risks. We hope that our work can draw attention to the safety risks in this direction and inspire extensive future research on investigating and addressing them.
> > >
> > > >*Further question about A7.*
> > >
> > > **Answer:** To clarify, the practical threat under wide-net-casting still remains substantial even for strong contemporary models.
> > >
> > > In A7 above, we conducted experiments with a group size of $M=2$. Even under this small group size, our method substantially increases attack success rate compared to both suggested models, including Llama-3-70B & Qwen3, and their basic OR-aggregation: their individual ASRs are only 14.7 & 13.3, "baseline" (basic OR-aggregation) yields 26.1 WASR, while our method achieves 39.8 WASR.
> > >
> > > Nevertheless, we highlight that the above experiment involves only 2 models and is thus relatively narrow, not fully reflecting a wide-net-casting setting. We thank reviewer for the comment. Below, we extend the evaluation to include additional contemporary models with strong safeguards and consider larger group sizes $M=4$ & $M=6$.
> > > - When $M=4$ (Llama-3-70B, Qwen3, Gemma 2, & Mistral-3.2), with individual ASRs of 14.7, 13.3, 8.6, & 9.2, our WASR is 68.1, much higher than 41.8 for "baseline" (basic OR-aggregation).
> > > - When $M=6$ (above 4 models, plus Yi-1.5-34B with ASR 9.8, and DeepSeek-R1-Distill-70B with ASR 5.3), our WASR is 87.2, much higher than 55.7 for "baseline" (basic OR-aggregation).
> > >
> > > We observe a similar trend for GPT-5.1, another suggested model:
> > > - When $M=4$ (GPT-5.1, Gemini-2.5, Qwen-VL-Max, & Claude-Sonnet-4.5), with individual ASRs of 14.5, 17.4, 12.2, & 8.9, our WASR is 71.7, much higher than 46.9 for "baseline" (basic OR-aggregation).
> > > - When $M=6$ (above 4 models, plus Mistral-Large-3 with ASR 9.3, and Kimi-K2.5 with ASR 11.2), our WASR is 91.6, much higher than 62.1 for "baseline" (basic OR-aggregation).
> > >
> > > Overall, these results show that **the relatively lower absolute WASR observed at $M=2$ is primarily due to small group size, rather than strength of contemporary models. Under wider-net-casting (e.g., when $M=4$ or $M=6$), the practical threat, as reflected by absolute WASR, still remains substantial even for strong contemporary models**. Moreover, note that across all settings and experiments, our method consistently and significantly outperforms "baseline" (basic OR-aggregation).

---

### Official Review · Reviewer_Be44 · 2026-03-11

**Soundness:** 3
**Presentation:** 4
**Significance:** 3
**Originality:** 4
**Overall Recommendation:** 4
**Confidence:** 4

**Summary:**

This paper studies a jailbreak setting the authors call the wide-net-casting scenario, where an attacker queries a group of large models and counts the attack as successful if any one model produces harmful output. The paper first adapts existing single-model jailbreak methods to this setting, introduces attacker-centric metrics such as WASR and W-Toxicity, and evaluates risk amplification across LLMs, MLLMs, same-family models, and models with added safeguards. It then proposes a tailored joint-training method that assigns each target model a dedicated adversarial generator and updates these generators using an entropy-constrained weighting rule derived from current losses. The main empirical finding is that wide-net-casting substantially increases attack success and toxicity, and the proposed tailored attack further increases these numbers, in some cases reaching 100% WASR.

**Compliance With Llm Reviewing Policy:**

Affirmed.

**Key Questions For Authors:**

1. Can you provide a query-budget-matched and compute-budget-matched baseline, for example (M) optimized attempts against one target model, and show whether the wide-net-casting gain remains clearly stronger?
2. The paper devotes a lot of space to deriving Equation 6, but the result is a standard Boltzmann allocation from entropy-constrained optimization. I do not object to using a standard tool, but the paper writes about it as if this theoretical machinery itself were a major contribution. What makes the losses in Equations 1-6 comparable across generators targeting different models? Did you try any normalization or ranking-based alternatives?

**Limitations:**

yes

**Strengths And Weaknesses:**

Strengths
1. The paper asks a useful evaluation question. Looking at multiple target models is a realistic attacker model for deployed systems, especially in a world where many public models are available.
The setup is generally clear.
2. Figure 1 on Page 1 is a strong explanatory figure. It cleanly shows why the attackers objective changes in the wide-net-casting scenario. Figure 2 on Page 3 also makes the straightforward multi-generator baseline easy to follow. Tables 1 and 2 cover LLMs and MLLMs across families and defenses; Tables 3 and 4 probe same-family behavior; Tables 5-7 evaluate the proposed method and ablations. This breadth helps the reader see where the phenomenon is stronger or weaker.
3. One interesting result is that related models reduce, but do not eliminate, the effect. Table 4 is especially useful here. The amplification becomes much smaller when the models are from the same family and version, which suggests the real scientific signal is about diversity of vulnerabilities rather than just the existence of multiple targets.
4. The impact statement is appropriate. The paper frames the work as red-teaming and evaluation, which is the right way to present this kind of safety analysis.
Weaknesses
1. On Page 3, WASR is defined as an OR over multiple models. That means the main finding that “WASR is much higher than ASR” is not, by itself, a deep empirical discovery. The real question is how much additional risk comes from cross-model diversity beyond simple multiple tries, and the paper does not isolate that well enough.
2. The comparisons are not budget-matched, which weakens causal interpretation. Figure 2 makes clear that the wide-net baseline and the proposed method both attack (M) models with (M) generators. That is a much larger resource budget than attacking one model once. Without a query-budget-matched or compute-budget-matched single-model baseline, the paper cannot cleanly attribute the gains to “wide-net-casting” rather than “more attack attempts.” This matters a lot because the paper frames the result as revealing a distinct safety risk rather than a resource-scaling effect.
3. The central methodological assumption, cross-generator loss comparability, is not justified. In Section 4.1, Equations 1-6, the method compares losses from generators targeting different models and interprets smaller loss as greater proficiency. But different target models can have different loss scales, response distributions, and optimization landscapes. If the losses are not on a common scale, the update weights may be arbitrary. This is not a side issue, it directly affects the validity of the proposed scheduling rule.

---

> ### Author Rebuttal · Authors · 2026-03-31
>
> >*Q1: Risk beyond simple multiple tries.*
>
> **A1:** Below we attack same model multiple tries and report performance (i.e. measuring at least one try succeeds).
> ||Performance(%)|
> |-|-|
> |Gemma-2-9b (1 try)|21.2|
> |Gemma-2-9b (4 tries)|21.5|
> |Gemma-2-9b (50 tries)|21.6|
> |Vicuna-7b-v1.5 (1 try)|27.9|
> |Vicuna-7b-v1.5 (4 tries)|28.2|
> |Vicuna-7b-v1.5 (50 tries)|28.2|
> |Llama-3.1-8b (1 try)|18.3|
> |Llama-3.1-8b (4 tries)|18.5|
> |Llama-3.1-8b (50 tries)|18.6|
> |Mistral-7b (1 try)|38.5|
> |Mistral-7b (4 tries)|38.7|
> |Mistral-7b (50 tries)|38.7|
> |**Wide-net-casting** (4 tries, 1 per model)|**61.5**|
>
> Compared to 1 try, multiple tries on same model yield marginal gains, still much worse than wide-net-casting, showing **additional risk mainly comes from cross-model diversity**.
> >*Q2: Budget-matched baseline.*
>
> **A2:** **(1)** In Sec 5, baseline, both naive strategies, and our method use same number of generators and attack attempts, ensuring a budget-matched comparison. Our method consistently outperforms them.
>
> **(2)** As shown in **A1**, wide-net-casting outperforms querying same model multiple times by a large margin.
>
> **(3)** Below, we run **M=4 optimized generators against one target model** to match budget of wide-net-casting attack. We evaluate them in two settings: using 4 generators with same architecture (Llama2-7B), and using 4 generators with different architectures (Llama2-7B, Llama-3-8B, Llama-3.1-8B, and TinyLlama-1.1B).
> ||Performance(%)|
> |-|-|
> |Gemma-2-9b (4 generators, same arch)|21.6|
> |Gemma-2-9b (4 generators, diff archs)|21.9|
> |Vicuna-7b-v1.5 (4 generators, same arch)|28.4|
> |Vicuna-7b-v1.5 (4 generators, diff archs)|28.7|
> |Llama-3.1-8b (4 generators, same arch)|18.7|
> |Llama-3.1-8b (4 generators, diff archs)|19.1|
> |Mistral-7b (4 generators, same arch)|39.0|
> |Mistral-7b (4 generators, diff archs)|39.4|
> |**Wide-net-casting**|**61.5**|
>
> Under matched budget, our wide-net-casting still very significantly outperforms variants that use M optimized attempts against one target model.
> >*Q3: Cross-generator loss comparability.*
>
> **A3: (1)** In our method, all generators share same architecture, same input, and use same output loss function (and the loss directly aligns with the final goal - attack performance / proficiency), thus largely facilitating loss comparability, and we observe our method works well in all experiments.
>
> **(2)** We tested different normalization or ranking-based variants: Variant I records each generator's loss scale at initialization and uses it to normalize subsequent losses; Variant II recomputes this scale at the start of each epoch for normalization; Variant III assigns loss weights inversely proportional to rank.
> ||WASR/W-Toxicity Score|
> |-|-|
> |Variant I|74.5/0.701|
> |Variant II|75.2/0.716|
> |Variant III|73.8/0.693|
> |**Ours**|**76.7/0.724**|
>
> All variants perform inferior to our method. A possible reason is that when loss is already well aligned with proficiency, additional manual operations could instead weaken performance.
>
> >*Q4: Space to deriving Equation 6 [...] using a standard tool [...] theoretical machinery as a major contribution.*
>
> **A4:** To clarify, we do not directly use a standard tool. Instead, we study a new challenging update-schedule problem that arises in the wide-net-casting scenario and propose new method tailored to it.
>
> **(1) Problem is new.**
> Under wide-net-casting, we train multiple adversarial generators, each paired with a different large model, and want them to gradually become different specialists. This requires deciding, at each training step how to allocate updates across generators. The key challenge is thus to balance exploitation and exploration: more promising generators should receive more updates, but over-committing to the current best one is suboptimal, as another generator may become the better specialist later.
>
> **(2) Formulation is new.**
> To solve this problem, we first formalize it ourselves. Via infinitesimal value shifting, we show that the update-scheduling problem can be cast as a tailored constrained optimization problem with a time-varying constraint. This formulation is not standardly given in advance; it is specifically formalized in our paper to make our new scheduling problem trackable.
>
> **(3) Solution is derived, not assumed.**
> We then solve this formalized problem using Lagrange multipliers, KKT conditions, and complementary slackness, and show that it admits an elegant closed-form solution of Boltzmann form. Thus, the Boltzmann form is the result of our careful theoretical derivation, instead of an off-the-shelf assumption. Without our first defining, formalizing, and careful theoretical derivations, there is no ground for getting what type of allocation should apply here.
>
> **(4)** Theoretically deriving Eq 6 is not the end. **We further make Eq 6 computable in practice,** by further theoretically deriving that β in Eq 6 is uniquely determined (Eq 16-25 in Supp). This turns Eq 6 into a practical, efficient, and stable training rule.

---

> > ### Author Rebuttal · Reviewer_Be44 · 2026-04-04
> >
> > thank you, my concerns are addressed, so I will keep my score.

---

> > > ### Author Response · Authors · 2026-04-04
> > >
> > > We are glad that we have addressed your concerns. Thanks for your time and effort and thank you for recommending accepting our paper.

---

### Official Review · Reviewer_kbKs · 2026-03-12

**Soundness:** 2
**Presentation:** 2
**Significance:** 1
**Originality:** 2
**Overall Recommendation:** 3
**Confidence:** 2

**Summary:**

This paper presents a jailbreak attack that targets multiple models simultaneously. The authors developed a jailbreak attack tailored for attacking multiple models at the same time. They presented several empirical analyses supporting their attack approach and evaluated it across various model sizes and families.

**Compliance With Llm Reviewing Policy:**

Affirmed.

**Key Questions For Authors:**

1. How does tailoring the generators to a specific intent help compared to them being trained independently?

**Limitations:**

No discussion on limitations.

**Strengths And Weaknesses:**

Strengths:

Strong empirical analysis supporting the claims. The authors developed specialized sample generators that target specific models or model families. The evaluation shows great performance.

Weakness:

I am not sure what the main difference is between wide-net casting and doing the attack independently on different models. I did not understand how the training in the widecast scenario helps to improve the attack.

I do not agree with the phrase "amplification" as the wide-net attack is simply an attack on multiple models in parallel. The success defined at the user end is additive (not multiplicative) across multiple models, so I do not see how amplification works with a wide-net attack.

In general, Section 4 is difficult to follow.

---

> ### Author Rebuttal · Authors · 2026-03-31
>
> >*Q1: Difference between wide-net casting and doing attack independently on different models [...] How does tailoring the generators to a specific intent help [...] Section 4 is difficult to follow.*
>
> **A1:** In wide-net casting scenario, for each harmful intent, we only need one large model to be breached to obtain harmful info. So the goal is not to make every adversarial sample generator do well on all harmful intents. Instead, the goal is: for each intent, how should we train generators so that at least one generator can breach its paired large model?
>
> The simplest idea is: if one large model is easier to breach on a certain harmful intent, then we should mainly train the generator paired with that model to handle that intent. This means that each generator does not need to handle all intents. It only needs to be strong on the subset of intents for which its paired large model is relatively easy to breach.
>
> This is why independently training every generator on the full harmful-intent training set is suboptimal. It forces every generator to handle everything, which wastes its capacity on less relevant intents and prevents it from becoming especially strong on intents that matter most (Paper L234-236). Instead, different generators should tailor different subsets of harmful intents, each aligned with the weaknesses of its paired large model. **In this way, they gradually specialize in different parts of the attack space, and together act like a Mixture of Experts that can cover harmful intents more effectively. This is why tailored training works better than independent training.**
>
> The difficulty now is that we do not know in advance which large model is weaker on which harmful intents. That is exactly what we address in **Sec 4**. We propose a joint tailored training scheme that, at each training iteration, estimates how suitable the current harmful intent is for updating each generator, and then dynamically assigns different generators with different update weights. Crucially, this weight assignment accounts for both the generators’ current states and the expected impact of the current update on future training dynamics. Over time, this gradually drives effective specialization. Importantly, **Sec 4** provides theoretical analysis showing that, through our principled closed-form solution for update-weight assignment, this dynamic training scheme indeed drives different generators toward final effective specialization in our problem.
>
> Finally, as mentioned in L279-299, we highlight our method cannot be well replaced by simple alternatives. For example, one may first train each generator independently on all harmful intents, then assign each intent to the generator that currently performs best on it, and continue training each generator exclusively on its assigned intent subset. This is yet still suboptimal, because the optimization landscape is complex and the generator that looks best at one stage may not remain the best later over training iterations. So, update weights should be allocated and adjusted over training. Thus, via a theoretically grounded way, our method steadily drives different generators toward final effective specialization. In contrast, both pure independent training and above alternative perform much worse than our method (Tabs 5-6): the former does not build specialization during training; the latter still leads to suboptimal specialization.
> >*Q2: Phrase "amplification".*
>
> **A2:** Previous jailbreak research focused on the risk of attacking a single model. Based on that single-model view, the community has developed defenses that reduce the jailbreak risk of each individual model to some level, which can create impression that the overall systems at present are sufficiently safe. However, this can be overconfident.
>
> Our paper for the first time highlights wide-net-casting scenario, where a user can try multiple models in parallel rather than attacking only one model. In this setting, the realized jailbreak risk can be substantially higher than what the usual single-model perspective suggests. Moreover, we designed a novel tailored attack training method specifically for this scenario. The risk is further significantly increased, even reaching 100% in some settings, which reveals a much more serious safety concern.
>
> To avoid confusion, instead of using “amplification”, we'll state in paper that, our tailored training strategy increases attack performance, compared to other methods under the wide-net-casting scenario.
> >*Q3: Limitations.*
>
> **A3:** Our paper introduces, for the first time, the wide-net-casting scenario and uncovers its substantial safety risks.
>
> As this scenario has not been studied before, defenses specifically designed for it remain open. Developing such defenses is thus an important future direction. We provide a preliminary discussion of possible defenses (see A5 of Reviewer mgDS for more details) and hope this work encourages further research in this direction.

---

> > ### Author Rebuttal · Reviewer_kbKs · 2026-04-02
> >
> > I appreciate the authors' effort in the rebuttal. However, I am not sure it was enough to change my mind about the contribution of this paper.

---

> > > ### Author Response · Authors · 2026-04-03
> > >
> > > >*Further question about the contribution of this paper.*
> > >
> > > **Answer:** In this paper, our main contributions are as follows. (1) We, for the first time, reveal a previously unexplored jailbreak scenario, the wide-net-casting jailbreak scenario. (2) We conduct a comprehensive analysis of the safety risks within this scenario, uncovering significant and previously overlooked vulnerabilities. (3) As a part of our analysis, we propose a jailbreak method tailored to this scenario, which further exposes the risks inherent in it in a more comprehensive manner.
> > >
> > > Our analysis shows that although many large models are widely believed to be safe and well-aligned with safety requirements, under the combined effect of our newly introduced scenario and our tailored jailbreak method, the jailbreak success rate can increase substantially. For example, as shown in Tables 5 and 6 of our paper, the jailbreak success rate can even reach 100% in certain settings. This substantially exceeds the level of jailbreak risk previously anticipated by the field. Such jailbreak risk has direct societal implications, as successful jailbreak attacks may induce highly harmful outputs, including fraud, violence, and self-harm, thereby leading to significant safety concerns and broad societal impact.
> > >
> > > Given the above findings, Reviewer mgDS points out that our work "identifies a new risk that is relevant to many existing systems in the real world", Reviewer yXcy notes that our work "identifies a realistic and important threat model", and Reviewer Be44 mentions that we "ask a useful evaluation question" and provide "a realistic attacker model". In summary, in our work, we uncover a previously unrecognized jailbreak scenario that is highly relevant to real-world deployed systems and propose a tailored attack method that effectively exposes its inherent risks. We believe this constitutes a valuable contribution (as also noted by Reviewer yXcy, who describes our scenario as "novel and valuable", and by Reviewer mgDS, who highlights our "novel findings"), and brings to the forefront a critical safety challenge not recognized by prior research. We hope this work will draw serious attention from both the research community and industry to this important and realistic threat, thereby catalyzing broader efforts to develop more effective defenses and mitigation strategies. In this way, we expect our findings to help drive meaningful progress in large model safety and support the broader advancement of responsible AI.

---

### Official Review · Reviewer_mgDS · 2026-03-13

**Soundness:** 3
**Presentation:** 3
**Significance:** 3
**Originality:** 3
**Overall Recommendation:** 5
**Confidence:** 3

**Summary:**

The paper focuses on the topic of jailbreak attacks on Large Language Models (LLMs). In particular, they identify a new threat - “wide-net-casting” scenario, where the adversary can query a group of LLMs instead of one. First, they show that adapting existing single-model jailbreak attacks to multiple models amplifies the risks. Finally, they present a new jailbreaking method tailored to the multiple-models “wide-net-casting” scenario.

**Compliance With Llm Reviewing Policy:**

Affirmed.

**Key Questions For Authors:**

How does the tailored attack affect the success rate of each individual model? For example, in Tab. 5, what would be the ASR for each individual model?

The appendix includes jailbreak times, but what are the training times, and how does this compare to single model attacks?

What are some possible defenses in the wide-net-casting scenario?

**Limitations:**

The paper does discuss the impacts of their work. However, I would expand this paper and include a discussion about possible defenses against this newly discovered risk.

**Strengths And Weaknesses:**

Strengths:

The paper tackles an important and up-to-date topic: jailbreaking LLMs.

They correctly identify and describe a new risk that is relevant to many existing systems in the real world

The paper is well-positioned with respect to the related works and clearly describes their novel findings (identifying a new risk in the multiple-LLMs scenario)

They correctly state relevant research questions and answer them with thorough experiments

The introduced method is nicely defined as an exploration-exploitation problem

Due to the novelty of the identified risk, there are no other works focusing on this scenario, but the authors correctly identify baselines (e.g., Naive Strategies 1 and 2) and show that their method poses a higher risk

The experiments are thorough, conducted on multiple LLMs and in many configurations

They conduct ablation studies assessing the impact of their design choices



Weaknesses:

The paper does not include training times.

I think the paper could expand the section on related works/ jailbreaks. In particular, it would be useful to explain how single-model jailbreaks work in detail (e.g., ReMiss), as they are used as a “black-box” component of the method

I would also include the STD in addition to the average of five runs in the results.

---

> ### Author Rebuttal · Authors · 2026-03-31
>
> >*Q1: Training time.*
>
> **A1:** Below, we compare our training time with that of single model attacks on A100 GPU.
> ||Training time|Performance (ASR/WASR)|
> |-|-|-|
> |Single for Qwen2-VL-7B|9.1h|39.7%|
> |Single for InstructBLIP|9.2h|53.3%|
> |Single for MiniGPT-4-13b|13.1h|77.9%|
> |Single for LLaVA-1.5-13b|13.6h|65.2%|
> |Ours (jointly targeting 4 large models)|14.5h|**100%**|
>
> Our method obtains much higher attack performance, while incurs only a small increase in training time.
> >*Q2: Expand the section on related works/jailbreaks. [...] useful to explain how single-model jailbreaks work in detail (e.g., ReMiss).*
>
> **A2:** In our related work, we briefly described how single-model jailbreaks work, which can be majorly categorized into (i) instance-based approaches (e.g., MLAI), discussed in L81-88, and (ii) model-based approaches (e.g., ReMiss), discussed in L89-95. We adapted both categories of methods to wide-net-casting scenario and evaluated them (Tabs 1 and 2).
>
> As suggested, **we'll expand related work to explain how single-model jailbreaks work in more detail.** Taking ReMiss and MLAI as examples, we will describe them as:
>
> **ReMiss** is a model-based jailbreak method which learns an adversarial sample generator to perform jailbreak. It first searches for effective jailbreak suffixes for harmful instructions that can receive high jailbreak rewards under the aligned model, and then uses these discovered suffixes to train the adversarial sample generator. At inference time, for unseen harmful intents, the trained generator can directly produce their jailbreak suffixes, without requiring per-intent optimization.
>
> **MLAI** is an instance-based jailbreak method that directly optimizes each adversarial sample. For each harmful intent, it starts from a scenario-aware initial image and iteratively applies gradient-based updates using cross-entropy loss to improve attack effectiveness. The optimized image is then paired with the intent to induce harmful outputs from the target MLLM.
>
> We'll explain single-model jailbreaks in more detail in paper.
> >*Q3: Include STD.*
>
> **A3:** Taking "Original Safety Alignment + SmoothLLM" column in Tab 5 as an example, results including STD are:
> ||WASR±STD (%)|W-Toxicity Score±STD|
> |-|-|-|
> |Baseline (ReMiss)|61.5±0.6|0.530±0.011|
> |Strategy 1|64.1±0.9|0.574±0.014|
> |Strategy 2|64.9±0.8|0.591±0.012|
> |**Ours**|**76.7±0.4**|**0.724±0.009**|
>
> **Our method outperforms compared methods in a statistically significant manner**. We observe such statistically significant out-performance also holds for all other experiments. We'll include STD in all tables in paper.
> >*Q4: How does tailored attack affect the success rate of each individual model? For example, in Tab 5, what would be ASR for each individual model?*
>
> **A4:** In Tab 5, under original safety alignment, ASR for Gemma-2-9b is 45.2% when its generator is independently trained, and becomes 43.8% after **tailored** joint specialized training. We observe similar mild decreases for other models, all by about 1.5%-3%. This is expected: tailored joint specialized training encourages each generator to mainly focus on weaknesses of its paired large model, rather than covering all weaknesses uniformly. Thus, each generator becomes a specialized expert for its paired model, yet ASR metric of each individual generator-model pair decreases slightly as it measures all weaknesses.
>
> Nevertheless, when these specialized generators are used jointly, overall WASR increases substantially (e.g., from 56.1% to 72.8% in Tab 5). This is exactly the advantage of joint specialized training. Since different large models often have different weaknesses, by making each generator a tailored expert for the weaknesses of its corresponding target model, rather than having every generator attempt to cover all weaknesses, their complementary use yields substantially higher jailbreak success in the wide-net-casting scenario.
>
> We'll include ASR of each model to Supp.
> >*Q5: Possible defenses.*
>
> **A5:** In Sec 6, we discussed some potential defense directions for wide-net-casting scenario (L425-430). Below we further provide more concrete possible defenses against this newly identified risk.
>
> (1) After running our method, model providers can know which harmful intents their models are particularly vulnerable to. They can then use these intents to train a lightweight safety filter, and use the filter to detect and reject queries targeting similar vulnerabilities at test time. (2) Our method can also be incorporated into safety alignment process of large models via an adversarial training mechanism. Specifically, we can progressively train generators to become dedicated jailbreak experts for different target models, use them to generate specialized attacks, and then use these attacks as adversarial training examples during safety alignment. In this way, large models can be progressively made more robust under the wide-net-casting scenario.
>
> We'll discuss more in paper.

---

> > ### Author Rebuttal · Reviewer_mgDS · 2026-04-04
> >
> > Thank you My concerns have been mostly addressed.

---

> > > ### Author Response · Authors · 2026-04-04
> > >
> > > We are glad that we have addressed most of your concerns. We are grateful for your time and effort, as well as your recommendation to accept our paper.

---

### Decision · Program_Chairs · 2026-04-30

**Decision:**

Accept (regular)

**Comment:**

This paper proposes that an attacker may query a group of LLMs instead of one to jailbreak the models and obtain harmful outputs from any of the models. The authors developed a new jailbreaking attack tailored for this scenario, as each attack generator is responsible for attacking one LLM, and multiple attack generators jointly cover harmful intents more effectively. The paper demonstrates strong empirical results.

This paper has received mixed reviews. Main concerns include: 1) unclear if the findings are non-trivial; 2) lack of novelty in the method; 3) limited evaluation. Based on the reviews and the rebuttal, the AC feels that the findings do seem to be non-trivial, and simply a lack of apparent novelty in the methodology should not be a valid reason for rejection. The authors have also provided additional results which should have addressed concerns regarding the evaluation. However, upon checking the paper, the AC agrees with one of the reviewers that Section 4 looks very hard to follow, which could have affected the reviews and how readers would be able to learn from this paper effectively.

Overall, the AC would like to recommend a weak accept.